# A Novel MEMS Gyroscope In-Self Calibration Approach

**DOI:** 10.3390/s20185430

**Published:** 2020-09-22

**Authors:** Qifan Zhou, Guizhen Yu, Huazhi Li, Na Zhang

**Affiliations:** School of Transportation Science and Engineering, Beihang University, Beijing 100191, China; yugz@buaa.edu.cn (G.Y.); huazhili@buaa.edu.cn (H.L.); b19028@buaa.edu.cn (N.Z.)

**Keywords:** in-self calibration, hand-held, Kalman filter, no support of external equipment

## Abstract

This paper presents a novel approach for hand-held low-cost MEMS (micro-electro-mechanical system) gyroscope in-self calibration. This method does not need the support of external high-precision equipment compared with traditional calibration scheme and can be accomplished by user hand rotation. In this approach, Kalman filter is designed to perform the calibration procedure and estimate gyroscope bias error, scale factor error and non-orthogonal error. The system observability is analyzed and the dynamic rotating conditions under which the sensor errors become observable are derived. The design principles of optimal calibration procedure are provided as well. Both simulated and practical experiments are carried out to test the validation of the proposed calibration algorithm. The achieved results demonstrate that the introduced approach can provide promising calibration scheme for the low-cost MEMS gyroscope.

## 1. Introduction

Rapid development of MEMS (micro-electro-mechanical system) technology and advances in manufacturing industry has made it possible to produce low-cost consumer-grade inertial sensors (i.e., gyroscope, accelerometer). The MEMS inertial sensor is advantageous in chip-size minimization, low-cost manufacturing, lower power consumption, and has been applied in multiple applications, such as vehicle and pedestrian navigation, wearable electronic devices, augmented reality (AR). However, the MEMS inertial sensor suffers from various error sources (i.e., bias, scale factor error, non-orthogonal error, random noise) and thermal drifts, which can cause negative effects on its utilization. Hence, a calibration process is highly required to determine the sensor errors and mitigate the drift. 

Calibration is defined as the process of comparing instrument outputs with known reference information to determine coefficients that force the output to agree with the reference information across the desired range of output values. A calibration experiment is conducted to compute deterministic errors of sensors in laboratory [1], and the calibrated parameters are employed to remove the sensor errors and derive a more reliable measurement. The calibration approaches and technologies have been well researched and studied, such as the local level frame (LLF) method and six-position static method [2]. However, these traditional calibration methods are primarily designed for in-lab tests and high-quality sensors, such as navigation or tactical grade IMUs (inertial measurement unit), and often require the use of special references (i.e., Earth rotation) such as alignment to a given frame or specialized equipment [3]. Therefore, using the specific high-level laboratory conditions to calibrate the low-cost MEMS sensor is always costly and meaningless.

Several researches or studies have been proposed to simplify the calibration procedure for MEMS inertial sensors. Skog [4] made use the fact that the norm of the gyroscope output ideally equals to the magnitude of applied rotational velocity. Li [5] presented a calibration scheme based on the principle that the time derivative of a reference vector can be expressed as the cross product of the angular velocity and the reference vector itself. Nieminen [6] utilized the standard multi-position method for consumer-grade IMUs’ calibration which also exploits the effect of the centripetal accelerations caused by the table rotation. Zhang [7] investigated the inter-triad misalignment between the gyroscope and accelerometer and the optimal calibration scheme design. The major advantages of these work are: (1) they do not need to use the Earth rotational velocity as the reference rotational velocity, which is weak signal (15°/h); (2) the calibration can be accomplished without aligning the IMU to the local level frame. However, they need the reference signals provided by the rotation table or other means to identify the gyroscope error.

Due to the inapplicability of using the external equipment (i.e., rotation table) in several practical fields, calibration schemes without the needs of expensive or cumbersome equipment are desirable and significant. Fong [8] developed an in-field hand-held calibration approach. This algorithm firstly generates multiple quasi-static stages, and compares attitude determined by the accelerometer outputs and the Strapdown inertial navigation algorithm with a cost function to calibrate the gyroscope. However, this approach needs the user to keep IMU stationary under different orientations to guarantee the accuracy. The static stage is judged by the threshold and the value selection is not discussed. You [9] introduced the pseudo-measurement concept and considered it as reference to perform the calibration. This method makes use of the constraint that the IMU’s position remains constant and velocity equals to zero when it is rotating strictly around its measurement center. The loosely-coupled integration navigation model was employed to estimate the inertial sensor biases and scale factor errors. Nevertheless, the pseudo-measurement (i.e., position, velocity), which is imposed as reference signal in the calibration, is weak correlated to sensor error; thus, the sensor error will not become well observable and realizable unless the IMU has experienced a continuous and suitable dynamic rotation. Moreover, this method is designed to use the gyroscope and the accelerometer triads of the IMU to calibrate each other, but the errors of both sensors are coupled together, and cannot be accurately estimated. The non-orthogonal error calibration is neglected in this work. Li [10] proposed a multi-level constrains autonomous gyroscope bias calibration algorithm. This approach consists of various constrains including pseudo-observation updates, the accelerometer and magnetometer measurement model, and magnetic disturbance detection, to contribute the bias estimation and is applied in pedestrian navigation scenario, while this approach only considers the sensor bias error and ignores the scale factor error and non-orthogonal error. These two kinds of error are proportional to the rotational velocity, which means that larger angular velocity introduces larger sensor error. Therefore, the estimated bias error is coupled with other error sources and became unreliable. Sachin [11] explored an in-package calibration approach and made use of a piezoelectric multi-modal mechanical stimuli stage to achieve the in-situ calibration. Umar [12] proposed in-field MEMS IMU calibration algorithm without support of external support; the rotation along vertical axis scheme is not considered and the sensor error’s observability is not analyzed. Farhad [13] provided an approach to use nonlinear model and transformed unscented Kalman filter (TUKF) to perform the low-cost MEMS IMU calibration procedure, but a rotation table is must needed. Additionally, the introduced gyroscope in-self calibration studies lack research work in analyzing the main error sources of MEMS sensor and providing theoretical calibration experiment design principles. Pablo [14] introduces a temperature-dependent calibration algorithm to calibrate triaxial sensors and develops an electromechanical prototype to perform the approach. Compared with this algorithm, our method does not need to consider the temperature element as the whole calibration procedure can be accomplished in short time without temperature variation and as such gyroscope bias error does not vary. Additionally, any other testing device is unnecessary. 

This paper proposes a novel low-cost MEMS gyroscope in-self calibration algorithm without the support of external equipment. Comparing with previous works, this method can identify the main error source of MEMS gyroscope, including bias, scale factor error, non-orthogonal error. We analyze different levels MEMS gyroscope error characteristics and the sensor measurement model and propose to calibrate the error under different dynamic conditions. Kalman filter is employed to design the calibration model, the system observability is analyzed, and the state vector observable conditions are determined. Thus, the IMU rotation scheme can be reasonably designed and implemented to calibrate the sensor errors in a quick and effective way. The main difference between the proposed algorithm and previous studied approaches are listed in Table 1.

The main advantages and contributions of the proposed method include: (1) It is an easy, convenient and novel approach to calibrate 9 gyroscope error parameters during short term (approximately 3–5 min); (2) The system observability is analyzed, and the sensor error observable conditions are obtained. Hence, the optimal calibration experiment can be reasonably and theoretically designed. (3) This approach does not require the support of external equipment and only needs to employ the pre-calibrated accelerometer output as reference signal. Moreover, it does not require other aiding information, such as GPS, magnetometer.

## 2. Main Error Sources of MEMS Gyroscope

Generally, gyroscope suffers from various error sources, such as bias, scale factor error, non-orthogonal error, acceleration errors (g-sensitivity), non-linearity, quantization error, and random errors [1]. Due to different manufacturing levels and chip production technologies, some kinds of sensor errors are dominant, while several others only have limited contribution. Therefore, it is critical to find out the main error sources of MEMS gyroscope and take them into consideration. We review the datasheet of different levels MEMS gyroscope (i.e., consumer level, high level), and summarize the sensor errors in Table 2, Table 3 and Table 4.

Table 2 lists the gyroscope characteristics of consumer-level (low level) IMU, MPU-9255/9150 [14,15]. Table 3 and Table 4 provide the gyroscope characteristics of high-level inertial measurement units ADI16367 [16] and IMU-440 [17]. 

According to the comparison of different levels MEMS gyroscope performance, the bias, scale factor error and non-orthogonal error are considered as the main error sources. The bias is at least 1°/s for the high-level MEMS gyroscope and is 5°/s for the consumer-level sensor. An uncompensated gyroscope bias in the sensor measurement will introduce an angle error (in roll or pitch) that is proportional to time *t*. Accordingly, an error proportional to $t^{2}$ and $t^{3}$ will be involved in velocity and position respectively. The bias error compensation determines the solution performance and is an essential part in navigation algorithm.

The scale factor error is at least 1%, which means that it introduces 1°/s error if the rotation velocity is 100°/s and the error will enlarge with the increase of angular velocity. The non-orthogonal error, which denotes that the three axes are not perfectly aligned to the coordinate and the rotation in one axis will be deduced in the other two axes, is approximately 1 degree. In this assumption, if the three axes rotation velocities are 100°/s, the non-orthogonal error will introduce 1.4 degrees measurement error to sensor output [3]. Hence, although the scale factor error and non-orthogonal error of high-level MEMS gyroscope is lower than that of consumer-level sensor, the measurement error caused by these errors is large and cannot be ignored. 

For the nonlinearity error, it is 0.1% and is the least error among MEMS gyroscope error characteristic. The effect of the nonlinearity error is much less than that caused by other errors. For instance, its error is only 0.1°/s when the rotation velocity is 100°/s. Therefore, for the calibration algorithm proposed in this paper, we only consider the bias, scale factor error and non-orthogonal error, and can safely ignore the nonlinearity error. 

In addition, this comparison also presents the effect of temperature on the gyroscope performance. For the consumer-level IMU, MPU-9255/9150, in the temperature range of −40 °C to +85 °C, the sensitivity scale factor variation is 4%, and the variation of Zero Rate Output (ZRO) is up to 20–30°/s. The scale factor error change of ADI1367 over temperature is 40 ppm/°C, and the bias variation with temperature is 0.01°/s/°C. The environmental conditions, such as the temperature variations and added voltage, will negatively impact the measurement performance of the MEMS gyroscope. Therefore, aiming to derive a more reliable navigation solution, it is more important to calibrate the MEMS gyroscope especially for the low-cost consumer-level, before usage in different application and environmental conditions.

## 3. Methodology

### 3.1. Sensor Error Model

Gyroscopes are angular rate sensors that provide either angular rate or attitude depending on whether they are of the rate sensing or rate integrating type. Measurements of angular rate can be modeled by the observation equation as follows:(1)$${\tilde{\omega }}_{ib}^{b}=\omega_{ib}^{b}+b_{g}+S_{g}\cdot \omega_{ib}^{b}+N_{g}\cdot \omega_{ib}^{b}+\varepsilon_{g}$$
where, ${\tilde{\omega }}_{ib}^{b}$ denotes gyroscope measurement vector. $\omega_{ib}^{b}$ is the true angular rate velocity vector. $b_{g}$ is the gyroscope instrument bias vector. $S_{g}$ is a matrix representing the gyroscope scale factor error. $N_{g}$ is a matrix representing non-orthogonal of the gyroscope triad. $\varepsilon_{g}$ is a vector representing the gyroscope sensor noise. 

The matrices $N_{g}$ and $S_{g}$ are given as: (2)$$N_{g}=\left[\begin{array}{ccc} 0 & G_{z} & -G_{y}\\ -G_{z} & 0 & G_{x}\\ G_{y} & -G_{x} & 0 \end{array}\right]\;\;\;\;\;\;\;\;\;\;\;\;\;\;S_{g}=\left[\begin{array}{ccc} S_{x} & 0 & 0\\ 0 & S_{y} & 0\\ 0 & 0 & S_{z} \end{array}\right]$$
where, $S_{\left(\cdot \right)}$ are the scale factors for the three gyroscopes and $G_{\left(\cdot \right)}$ represent the non-orthogonal error. 

### 3.2. Bias Calibration

The bias is defined as the signal output of the sensor when it is not experiencing any rotation. According to this definition and the measurement error model in Equation (1), when the sensor is kept in stationary, the true angular rate velocity is zero and the items related to $\omega_{ib}^{b}$ in Equation (1) disappear. Thus, the gyroscope output is the bias error.

The random noise $\varepsilon_{g}$ involved in the gyroscope measurement, can be modeled as zero-mean white noise. The bias error can be calculated by averaging the gyroscope measurement in static period.
(3)$$b_{g}=\frac{1}{N}\overset{N}{\underset{i=1}{\sum }}{\tilde{\omega }}_{ib}^{b}$$
where, *N* denotes the collected measurement points. Generally, we can collect 3–5 min gyroscope measurement in static period and the bias error is calculated as the average.

### 3.3. Scale Factor and Non-Orthogonal Calibration

With given initial attitude derived from alignment procedure, the gyroscope measurement can be integrated to calculate the orientation information through Strapdown inertial navigation algorithm. However, the computed attitude will drift over time and the error is gradually accumulated because of the sensor error. In stationary or low dynamic condition, the accelerometer output can be used to estimate the orientation relative to horizontal plane (i.e., pitch and roll). The attitude derived from accelerometer output is independent in different time epochs and not affected by accumulated error. Hence, based on the different sensors’ complimentary error propagation characteristics, we can make use of the accelerometer-derived attitude as reference signal to evaluate the attitude error introduced during integration process, and consequently determine the gyroscope error.

During the calibration process, the IMU is handheld by user and rotated along its axes slowly to avoid introducing external acceleration. The IMU orientation keeps varying during this procedure and the attitudes derived from different inertial sensors are compared to amend the attitude error and determine the sensor errors. A Kalman filter [18] is designed to estimate the scale factor and non-orthogonal errors of gyroscope. The attitude error propagation equation, which includes sensor error, is utilized as the system dynamic model. The relationship between the accelerometer output and attitude error is modeled as the measurement equation.

#### 3.3.1. Dynamic Model

A simplified psi-angle error mode is applied as the dynamic model in Kalman filter, where the Psi-angle error represents the Euler angles between true navigation n frame (East-North-Up) and the platform frame (computed navigation frame $n_{c}$ frame). Adopting this kind of angle error mode is advantageous as it avoids non-linear problem and is easy to be implemented. The dynamic model is expressed as:(4)$$\dot{\varphi }=-\left[\omega_{in}^{n}\times \right]\varphi -\delta \omega_{in}^{n}+C_{b}^{n}\delta \omega_{ib}^{b}$$
where, φ denotes the attitude error expressed in *n* frame (i.e., navigation frame), $\omega_{in}^{n}$ denotes the *n* frame rotation angular rate vector relative to the inertial frame (*i* frame) expressed in *n* frame. $C_{b}^{n}$ denotes the Direction Cosine Matrix (DCM) from body frame (i.e., *b* frame) to *n* frame. The symbol “×” denotes cross product of two vectors. $\delta \omega_{ib}^{b}$ denotes the gyroscope output error. 

The items $\omega_{in}^{n}$, $\delta \omega_{in}^{n}$ in Equation (4) are:(5)$$\omega_{in}^{n}=\omega_{ie}^{n}+\omega_{en}^{n}=\left[\begin{array}{c} -\frac{V_{N}}{R_{M}+h}\\ \omega_{ie}cosL+\frac{V_{E}}{R_{M}+h}\\ \omega_{ie}sinL+\frac{V_{E}}{R_{N}+h}tanL \end{array}\right]$$
(6)$$\delta \omega_{in}^{n}=\left[\begin{array}{c} -\frac{\delta V_{N}}{R_{M}+h}+\delta h\frac{V_{N}}{R_{M}+h}\\ -\omega_{ie}sinL\delta L+\frac{\dot{\lambda }cosL}{R_{N}+h}\delta h+\frac{{\delta V}_{E}}{R_{M}+h}\\ \left(\omega_{ie}cosL+\frac{\dot{\lambda }}{cosL}\right)\delta L-\frac{\dot{\lambda }sinL}{R_{N}+h}\delta h+\frac{{\delta V}_{E}}{R_{N}+h}tanL \end{array}\right]$$


Because the hand-held rotation strategy is adopted to calibrate the sensor error, the velocity will be extremely low (approximately 0 m/s) and the position change is reasonable to be considered as zero. In addition, the velocity components and the position error are either divided by the Earth’s radius (or its square) or are multiplied by the Earth rotation. Hence, the two rotation items are nearly zero, which is much less than the sensor error item, and as such can be safely ignored. Hence, the Equation (4) is simplified as:(7)$$\dot{\varphi }=C_{b}^{n}\cdot \delta \omega_{ib}^{b}$$

This equation illustrates that the attitude error is caused by angular velocity errors in measuring the rotation between the two frames (i.e., ***i*** frame and ***n*** frame). Due to the bias error has been calibrated by averaging the gyroscope output in static period, only the scale factor error and non-orthogonal error are considered and the equation is rewritten as:(8)$$\dot{\varphi }=C_{b}^{n}\cdot \left(S_{g}+N_{g}\right)\delta \omega_{ib}^{b}$$

We model the scale factor error and non-orthogonal error in system dynamic model as random constant [19] and the dynamic model is described as:(9)$$\left[\begin{array}{c} \dot{\varphi }\\ {\dot{S}}_{g}\\ {\dot{N}}_{g} \end{array}\right]=\left[\begin{array}{ccc} O_{3\times 3} & C_{b}^{n}\cdot diag\left(\omega_{ib}^{b}\right) & C_{b}^{n}\cdot \left[\omega_{ib}^{b}\times \right]\\ O_{3\times 3} & O_{3\times 3} & O_{3\times 3}\\ O_{3\times 3} & O_{3\times 3} & O_{3\times 3} \end{array}\right]\left[\begin{array}{c} \varphi \\ S_{g}\\ N_{g} \end{array}\right]+\left[\begin{array}{c} C_{b}^{n}\\ O_{3\times 3}\\ O_{3\times 3} \end{array}\right]w$$
where, φ is 3 × 1 vector, which contains the pitch, roll and yaw angle errors δθ, δγ, δϕ. $O_{3\;\times \;3}$ denotes the 3 × 3 zero matrix. $\left[\omega_{ib}^{b}\times \right]$ denotes the skew-symmetric matrix of gyroscope measurement. $diag\left(\omega_{ib}^{b}\right)$ denotes the diagonal matrix with the gyroscope measurement as its diagonal elements and **w** denotes the process noise and is set according to the gyroscope Angular Random Walk (ARW) [20].

#### 3.3.2. Measurement Model

The acceleration residuals between the projection of accelerometer measurement in computed navigation frame $n_{c}$ and the local gravity acceleration are used to build the measurement model. The acceleration projection and gravity acceleration are written as:(10)$$g^{n}={\left[0\;\;0\;\;\;g\right]}^{T}\;f^{n_{c}}=C_{b}^{n_{c}}\cdot f^{b}\;f^{b}=C_{b_{a}}^{b_{g}}{\left[a_{x}a_{y}a_{z}\right]}^{T}$$
where, $g^{n}$ denotes the local gravity acceleration. $\left(a_{x},a_{y},a_{z}\right)$ denote the accelerometer measurements. $C_{b}^{n_{c}}$ denotes the transformation matrix between b frame and $n_{c}$ frame,$\;C_{b_{a}}^{b_{g}}$ denotes the misalignment between accelerometer and gyroscope traids. It should be noted that the accelerometer has been well calibrated and its biases, scale factor errors and non-orthogonal errors have been removed [21]. 

Followed by DCM chain rule, $C_{b}^{n_{c}}$ is expressed as:(11)$$C_{b}^{n_{c}}=C_{b}^{n_{c}}\cdot C_{b}^{n}\;C_{n}^{n_{c}}=I-\left[\varphi \times \right]$$
where, [φ×] denotes the skew matrix of attitude error.

The acceleration residual is the difference between $g^{n}$ and $f^{n_{c}}$. Substitute Equation (10) into the residual and we can derive.
(12)$$\delta f=g^{n}-f^{n_{c}}=g^{n}-C_{b}^{n_{c}}f^{b}\;=g^{n}-\left(I-\left[\varphi \times \right]\right)C_{b}^{n}f^{b}\;=g^{n}-\left(I-\left[\varphi \times \right]\right)f^{n}=\left[\varphi \times \right]f^{n}$$

Therefore, the measurement model is derived as:(13)$$Z=HX+v=\left[\begin{array}{cc} \begin{array}{ccc} 0 & g & 0\\ -g & 0 & 0\\ 0 & 0 & 0 \end{array} & O_{3\times 6} \end{array}\right]\left[\begin{array}{c} \varphi \\ S_{g}\\ N_{g} \end{array}\right]+v$$
where $O_{3\;\times \;6}$ denotes a 3 × 6 zero matrix. **H** denotes the design matrix and **v** denotes the measurement noise.

The system model adopts the attitude error propagation equation, which combines the attitude error and sensor error. This model establishes a direct relationship between these two errors and is beneficial to increase the parameter observability. Moreover, the attitude error model is expressed in linear form and the Kalman filter can provide an optimal solution. The acceleration residuals, instead of the pitch and roll errors, are used as the observation vector. Although, using the orientation difference from various sensors is an intuitive way to implement the observation vector, it will introduce the singularity problem in the attitude calculation when the pitch angle reaches ± 90°. However, the acceleration residuals include equivalent errors information of pitch and roll angles, without being affected by singularity problem. 

## 4. Observability Analysis

### 4.1. Observability Definition

Observability describes the ability of estimating the system states [22], which is a measure for how well system states can be inferred by knowledge of the measurement. Analyzing the gyroscope sensor error observability is beneficial to know the state vector under which dynamic condition (i.e., continuous rotation, kept static) becomes observable and well estimated; thus, the gyroscope calibration scheme can be further optimally designed and implemented.

Consider the linear system:(14)$$\dot{x}\left(t\right)=A\left(t\right)x\left(t\right)\;y\left(t\right)=C\left(t\right)x\left(t\right)$$
where, **A**(*t*) and **C**(*t*) are, the **n** × **n** and **p** × **n** matrices whose entries are continuous functions of time defined over (−∞,+∞), respectively.

The dynamic equation is observable at $t_{0}$ if there exists a finite time $t_{1}>t_{0}$ such that for any state $x_{0}$ at time $t_{0}$, the knowledge of the output y(t) over the time interval suffices to determine the state $x_{0}$.

Define a sequence of **p** × **n** observability matrix $N_{0}\left(t\right),N_{1}\left(t\right),\cdots N_{n-1}\left(t\right)$ by the equation:(15)$$N_{n}\left(t\right)=N_{n-1}\left(t\right)A\left(t\right)+\frac{d}{dt}N_{n-1}\left(t\right)\;\;n=0,1,2\;\cdots k\;N_{0}\left(t\right)=C\left(t\right)$$

Suppose **A**(*t*) and **C**(*t*) in the system are analytic functions of ***t***. Then the time-varying system **A**(*t*) and **C**(*t*) is observable at the time $t_{0}$ if there exists a finite time $t_{1}>t_{0}$ such that the rank of the matrix is *n*.
(16)$$W=\left[\begin{array}{c} N_{0}\left(t_{1}\right)\\ N_{1}\left(t_{1}\right)\\ \vdots \\ N_{n-1}\left(t_{1}\right) \end{array}\right]$$

The above observability tests are the same as finding a state vector **x**(*t*) such that
(17)$$y\left(t\right)=N_{0}\left(t\right)x\left(t\right)=0\;\dot{y}\left(t\right)=N_{1}\left(t\right)x\left(t\right)=0\;\vdots \;y^{n-1}\left(t\right)=N_{n-1}\left(t\right)x\left(t\right)=0$$

For a time $t_{1}>t_{0}$, if there is no nonzero state that satisfies the above conditions, then the system is observable at $t_{0}$. 

### 4.2. Observability Analysis

According to observability definition, and the system transition and design matrices, ***F***(t) and **H**, the system observability is determined by the rotational angular velocity **ω** and the IMU attitude (i.e., the transformation matrix $C_{b}^{n}$). By substituting (9) and (13) into (15), the proposed system observability matrix is described as:(18)$$N_{0}=H\;N_{1}=\;\left[\begin{array}{cc} O_{2\times 3} & \begin{array}{cccccc} gC_{21}\omega_{x} & gC_{22}\omega_{y} & gC_{23}\omega_{z} & N_{1_{17}} & N_{1_{18}} & N_{1_{19}}\\ -gC_{11}\omega_{x} & -gC_{12}\omega_{y} & gC_{13}\omega_{z} & N_{1_{27}} & N_{1_{28}} & N_{1_{29}} \end{array} \end{array}\right]\;N_{1_{17}}=\;g\left(C_{22}\omega_{z}-C_{23}\omega_{y}\right)\;\;N_{1_{18}}=\;-g\left(C_{21}\omega_{z}-C_{23}\omega_{x}\right)\;\;\;N_{1_{19}}=g\left(C_{22}\omega_{z}-C_{23}\omega_{y}\right)\;N_{1_{27}}=\;g\left(C_{12}\omega_{z}-C_{13}\omega_{y}\right)\;\;N_{1_{28}}=\;-g\left(C_{11}\omega_{z}-C_{13}\omega_{x}\right)\;\;\;N_{1_{19}}=-g\left(C_{11}\omega_{y}-C_{12}\omega_{x}\right)\;N_{2}={\dot{N}}_{1}\;\;N_{3}={\dot{N}}_{2}$$
where, $O_{2\;\times \;3}$ denotes 2 × 3 zero matrix. $C_{mn}$ denotes the element in $C_{b}^{n}$ matrix which is in the *m**^th^* row and *n**^th^* column. $C_{b}^{n}$ denotes the transformation matrix between *b* frame and *n* frame and is written as: (19)$$C_{b}^{n}=\left[\begin{array}{ccc} \cos\gamma \cos\emptyset -\sin\gamma \sin\emptyset \sin\theta & -\sin\emptyset \cos\theta & \sin\gamma \cos\emptyset +\cos\gamma \sin\emptyset \sin\theta \\ \cos\gamma \sin\emptyset +\sin\gamma \cos\emptyset \sin\theta & \cos\emptyset \cos\theta & \sin\gamma \cos\emptyset -\cos\gamma \cos\emptyset \sin\theta \\ -\sin\gamma \cos\emptyset & \sin\theta & \cos\gamma \cos\emptyset \end{array}\right]$$

The body frame (i.e., *b* frame) in this paper is defined as Right-Forward-Up and the navigation frame (i.e., *n* frame) is defined as East-North-Up (ENU). Three consecutive rotations from the *n* frame to *b* frame is ϕ along *z*-axis, θ along *x*-axis and γ along *x*, *y*, *z*-axes. Rotating the IMU in hand in a free-style will lead that the system process equations become time-varying and complicate the theoretical observability analysis process. Aiming to derive a reliable analysis and estimation result, we consider the typical or special rotation and analyze the system observability in these conditions. The state vector observability is investigated under two dynamic conditions: (1) Rotation along vertical axis, which means that the IMU rotates along its sensitive axis when this axis is vertical to horizontal plan; (2) Rotation along horizontal axis, which means that the IMU rotates along its sensitive axis when it is aligning the horizontal plane.

#### 4.2.1. Rotation along Vertical Axis

Assuming the IMU is put in flat platform with the *x*-axis and *y*-axis aligning to horizontal plane, and the *z*-axis is perpendicular to the plane. The rotation vector expressed in *b* frame is (0,0,ω). The roll and pitch angles are both zero. In this condition, the items $N_{2},N_{3}$…in observability matrix will become zero matrix and the observability matrix is written as:(20)$$W=\left[\begin{array}{c} N_{0}\\ N_{1} \end{array}\right]=\left[\begin{array}{ccccccccc} 0 & -g & 0 & 0 & 0 & 0 & 0 & 0 & 0\\ g & 0 & 0 & 0 & 0 & 0 & 0 & 0 & 0\\ 0 & 0 & 0 & 0 & 0 & 0 & g\cos\emptyset \cdot \omega_{z} & g\sin\emptyset \cdot \omega_{z} & 0\\ 0 & 0 & 0 & 0 & 0 & 0 & -g\sin\emptyset \cdot \omega_{z} & g\cos\emptyset \cdot \omega_{z} & 0 \end{array}\right]$$

Substitute $N_{0}\left(t\right),N_{1}\left(t\right)$ into Equation (17), and the observability test is described as:(21)$$-g\cdot \delta \gamma =0\;g\cdot \delta \theta =0\;g\cos\emptyset \cdot \omega_{z}\cdot G_{x}+g\sin\emptyset \cdot \omega_{z}\cdot G_{y}=0\;-g\sin\emptyset \cdot \omega_{z}\cdot G_{x}-g\cos\emptyset \cdot \omega_{z}\cdot G_{y}=0$$

Correspondingly, only in the case that the elements $\delta \gamma ,\;\delta \theta ,G_{x},G_{y}$ in state vector are zero, the condition listed in Equation (17) is satisfied. Hence, when the *z*-axis is vertical to the horizontal plane and the IMU is rotating with respect to it, the attitude error δγ, δθ and the non-orthogonal error $G_{x},G_{y}$ are observable. 

Furthermore, when the *y*-axis is perpendicular to the horizontal plane and the IMU rotates along this axis, the attitude error and the non-orthogonal error $G_{x},G_{z}$ are observable. If the IMU is rotating with respect to *x*-axis and this axis is vertical to the horizontal plane, the attitude error and the non-orthogonal error $G_{x},G_{z}$ are observable.

#### 4.2.2. Rotation along Horizontal Axis

Assuming the IMU is put in flat platform and the *z*-axis is perpendicular to the horizontal plane, the three axes rotation are separately (0,ω,0). The initial roll and pitch angles are both zero. In this condition, we ignore the $N_{2},N_{3}$… in observability matrix and the observability matrix is written as:(22)$$W=\left[\begin{array}{c} N_{0}\\ N_{1} \end{array}\right]=\left[\begin{array}{ccccccccc} 0 & -g & 0 & 0 & 0 & 0 & 0 & 0 & 0\\ g & 0 & 0 & 0 & 0 & 0 & 0 & 0 & 0\\ 0 & 0 & 0 & 0 & g\cos\emptyset \cos\theta \cdot \omega_{z} & 0 & g\cos\emptyset \sin\theta \cdot \omega_{z} & 0 & g\sin\emptyset \cdot \omega_{z}\\ 0 & 0 & 0 & 0 & g\sin\emptyset \cos\theta \cdot \omega_{z} & 0 & g\sin\emptyset \sin\theta \cdot \omega_{z} & 0 & g\cos\emptyset \cdot \omega_{z} \end{array}\right]$$

Consequently, the observability test is described as:(23)$$-g\cdot \delta \gamma =0\;g\cdot \delta \theta =0\;g\cos\emptyset \cos\theta \cdot \omega_{z}\cdot S_{x}+g\cos\emptyset \sin\theta \cdot \omega_{z}\cdot G_{y}+g\sin\emptyset \cdot \omega_{z}\cdot G_{z}=0\;g\sin\emptyset \cos\theta \cdot \omega_{z}\cdot S_{x}+g\sin\emptyset \sin\theta \cdot \omega_{z}\cdot G_{y}+g\cos\emptyset \cdot \omega_{z}\cdot G_{z}=0$$

The rank of observability matrix is 4, while five states are available in the observability test. The attitude error, δγ and δθ are not coupled with other elements, and these two states are always observable. The combination of the rest three items, *y*-axis scale factor $S_{y}$ and the non-orthogonal error $G_{x},G_{z}$ are observable. Because the non-orthogonal errors have become observable and can be well estimated when the IMU is rotating with respect to its vertical axis. Therefore, though the scale factor error is coupled with the non-orthogonal error, it still can be derived and estimated. 

Accordingly, in the same condition, if the IMU rotates with respect to the *x*-axis, the combination of $\left(S_{y},G_{y},G_{z}\right)$ is observable. If the IMU has an initial 90°or −90° roll angle that its *z*-axis is aligned to the horizontal plane, the combination of $\left(S_{z},G_{x},G_{y}\right)$ is observable when the IMU rotates alongthe *z*-axis. 

#### 4.2.3. Observability Analysis Summarization

Based on the theoretical analysis of system observability, the relationship between the gyroscope error observability and dynamic conditions are summarized in the Table 5.

Table 5 lists the observable states, orientations, and dynamic rotations. To summarize, the non-orthogonal error is observable when the IMU is rotating with respect to its vertical axis. The combination of three-sensor errors, including one scale factor error and two non-orthogonal errors, is observable when the IMU is rotating with respect to its horizontal axis. 

The scale factor and non-orthogonal errors are coupled together when the IMU is rotating along the horizontal plane; however, the non-orthogonal error can be well estimated when the IMU rotates with respect to the vertical axis. Therefore, in order to acquire a reliable error estimation result, the calibration procedure should be designed as rotating along the vertical axis to derive the three non-orthogonal errors first, and then rotating with respect to the horizontal axis to estimate the three scale factor errors.

## 5. Test and Results

Simulated and practical experiments have been carried out to test and verify the proposed calibration algorithm. In simulated experiment, we first design the sensor rotation sequence to generate true sensor data according to observability analysis and add the preset sensor error to data which is used to perform the calibration approach. In practical experiment, the rotation follows the same sequence as that performed in simulation. In each experiment, the converge of estimated scale factor error and non-orthogonal error and their corresponding element in P matrix are drawn, compared and analyzed. In simulation experiment, we compare the estimated sensor error and preset error parameter to derive the absolute error and relative error. In practical experiment, the attitude calculated by sensor data with and without error compensation are compared to demonstrate the validation of calibrated parameter and all the figures are plotted in Matlab.

### 5.1. Simulation Experiment

Aiming to derive a reliable gyroscope error calibration result, the IMU is designed to rotate along vertical axis first to estimate the non-orthogonal error and consequently along horizontal axis to estimate the scale factor error based on the system observability analysis. The rotation scheme design is described in the Table 6.

The IMU keeps static with zero attitude in the first 10 s and rotates with respect to *z*-axis in clockwise and counterclockwise directions. Then the IMU rotates 90 degrees along *y*-axis to move the *x*-axis vertical to horizontal plane and rotates with respect to *x*-axis. Consequently, the IMU separately rotates along *x*, *y*, *z*-axes in horizontal plane. Table 6 lists the designed rotation sequence including the time period, direction and rotating axis. 

The gyroscope error including scale factor error, non-orthogonal error, ARW, and the accelerometer error Velocity Random Walk (VRW) are simulated according to the consumer level MEMS sensor performance and listed in Table 7. These error parameters are substituted into the sensor measurement model to generate the output and are used for calibration. 

Figure 1 shows the simulated inertial sensor measurement and Figure 1a,b separately illustrate the rotation and acceleration. The blue, red and yellow lines denote the measurement in *x*, *y*, *z* axes. The proposed calibration algorithm is employed to estimate the sensor error and the initial state vector and variance-covariance matrix **P** are set as:(24)$$X={\left[\begin{array}{ccccccccc} 0 & 0 & 0 & 0 & 0 & 0 & 0 & 0 & 0 \end{array}\right]}^{T}\;P=\mathrm{diag}{\left(\begin{array}{ccccccccc} 3 & 3 & 3 & 3 & 0.05 & 0.05 & 0.05 & 0.05 & 0.05 \end{array}\right)}^{2}$$

The initial state is set as zero-vector due to the lack of sensor error prior knowledge. The **P** matrix is set based on the maximum sensor error. The error estimation result is drawn in Figure 2.

Figure 2 shows the sensor error estimation result, Figure 2a denotes the scale factor error and Figure 2b shows the non-orthogonal error result. The blue, red and yellow line denote the error of *x*, *y* and *z* axes. Figure 3 shows the elements of **P** matrix, Figure 3a,b denote the elements corresponding to the scale factor error and the non-orthogonal error; Figure 3c,d denote the elements corresponding to attitude error and heading error. The reason for investigating the elements of covariance matrix **P** is that if its diagonal element experiences a large decrease from its initial value, the corresponding state becomes observable and its observability is large [20].

As shown in Figure 3, because the IMU keeps static in the first 10 s, the state vector is not observable, and the diagonal element of P matrix does not change. Starting from 10 s, the IMU rotates with respect to *z*-axis, the $G_{x},G_{y}$ converge to the reasonable value and their corresponding **P** matrix elements decrease dramatically. When the IMU rotates along x axis vertically beginning from 40 s, the $G_{z}$ begins to converge and its corresponding **P** matrix element decreases. The non-orthogonal error estimation result verifies the observability analysis that rotating along vertical axis is able to make the non-orthogonal error observable and achieve a reliable calibration result.

For the scale factor error, the IMU rotates with respect to the *x*-axis and *z*-axis separately starting from 80 s and 126 s, then the $S_{x}$ and $S_{z}$ begin to converge and their corresponding **P** matrix elements experience a large decrease simultaneously. The scale factor error estimation result also demonstrates the observability analysis that rotating along the IMU horizontal axis is beneficial to make the scale factor error observable and estimate this error. 

For the attitude error (i.e., pitch error, roll error), the two parameters are observable in the filter and is unrelated to the system dynamic condition. Hence, the **P** matrix elements corresponding to these two errors converge after the filter begins to work. The heading error is unobservable during the process, because the system measurement vector is uncorrelated to the heading error and cannot provide helpful correction, which explains that the its corresponding P matrix element diverges in the calibration procedure. In addition, the bias, scale factor error and non-orthogonal error estimation results are listed in Table 8 and Table 9. The absolute error and relative error compared with true value are also provided. 

The gyroscope error calibration result is reasonable and reliable. The maximum relative error is 6.8%, and the minimum relative error is only 1%. Moreover, if the random noise of inertial sensor can be decreased in the simulated process, a more accurate result can be acquired. 

Additionally, we utilize the singular value decomposition (SVD) of the observability stripped observation matrix to evaluate the degree of observability of each state. We separately calculate the singular value of the observability matrix with and without the observable state according to analysis result. Table 10 and Table 11 give the result.

Table 10 and Table 11 list the singular value of the SOM when the IMU rotates under different dynamic motion. In order to analyze the single state observability, we separately calculate the singular values with full states (9 states) and chosen states (6 or 7 states) and make the comparison. The zero singular values are ignored and only the non-zero values are listed in the tables. 

As described in tables, two singular values 9.8 are available in each group. The reason is that the pitch error and roll error are always observable during the procedure, and 9.8 is their singular value. Hence, only need to analyze the other values. When the IMU is rotating with respect to the *z*-axis, the singular values of SOM (Stripped Observability Matrix) with the full states are 8.52 and 8.50. While the singular value become 0.61935 and 0.375294 when the states $G_{x}$ and $G_{y}$ are not considered in the observability matrix. It shows that the system observability degree decreases dramatically and only the attitude errors are observable in this condition. Meanwhile, it demonstrates that $G_{x}$ and $G_{y}$ are observable when the IMU rotates along the *z*-axis vertically. Based on this analysis and the comparison of singular values under various dynamic conditions, it comes to the same result with the observability analysis that the scale factor error is observable when the IMU rotates horizontally and the non-orthogonal error is observable when the IMU is rotating vertically. 

### 5.2. Practical Experiment

We employ the consumer level MESM sensor MPU-9150 in practical experiment to test the proposed calibration algorithm. The MPU-9150 is a 9-axis inertial sensor which includes 3-axis gyroscope, 3-axis accelerometer and 3-axis magnetometer. The rotation sequence follows the designed scheme described in simulation experiment that rotating along the vertical axis first and then along the horizontal axis. The collected data is illustrated in Figure 4.

Figure 4 illustrates the collected inertial data in practical experiment. Figure 4a,c show the gyroscope and accelerometer measurement. Figure 4b shows the gyroscope measurement of static period in the first 10s. The rotation data of static period is averaged to derive the sensor bias. The initial state vector and covariance matrix **P** is set the same as that in the simulated experiment. The sensor error calibration result is shown in Figure 5.

Figure 5 shows the sensor error estimation result, Figure 5a denotes the scale factor error and Figure 5b shows the non-orthogonal error result. The blue, red and green lines denote the errors of *x*, *y* and *z* axes. Figure 6 shows the element of **P** matrix, Figure 6a,b denote the elements corresponding to the scale factor error and non-orthogonal error; Figure 6c,d denote the elements corresponding to the attitude error and heading error.

Due to the IMU keeps static in the first 10 s, the sensor errors (i.e., scale factor error, non-orthogonal error) are not observable and cannot be well estimated. After the static period, the IMU starts to rotate along *z*-axis vertically, the $G_{x},G_{y}$ begin to converge and their corresponding P matrix diagonal elements experience large decrease. Starting from 30 s, the IMU rotates along *x*-axis vertically and its corresponding P matrix decreases. The IMU starts to rotate along *x*, *y*, *z* axis separately approximately since 52 s, 72 s, 90 s, and the scale factor errors, $S_{x},S_{y},S_{z}$ also begin to gradually converge from these time epochs. Meanwhile, their corresponding P matrix diagonal elements decrease. 

In order to test the validation of calibration algorithm, we rotate the IMU in free style and separately calculate the attitude using the sensor data with and without error compensation. The attitude is computed with the orientation integration algorithm and the results are compared with the reference. The attitude calculation results are illustrated in the following figures.

Figure 7 shows the attitude calculation result, Figure 7a,c,e show the roll, pitch, heading results and Figure 7b,d,f show the attitude results in the selected periods for better comparison. The blue line denotes the attitude result calculated by the inertial sensor data which has been compensated by the error parameters derived from proposed calibration approach. The red line denotes the attitude calculated by the data in which only the bias has been removed, and the green line denotes the reference. 

As shown in the right side three figures which have been enlarged in selected periods, the attitude with error compensation (blue line) is closer to the reference (green line) and the attitude without error compensation (red line) drifts away from the reference. The attitude comparison demonstrates the validation of gyroscope error estimation result because the attitude performance has been improved through well sensor error compensation, while the attitude without error compensation has a poorer performance. The attitude error result and its statistical result are shown in Figure 8 and Table 12.

Figure 8 shows the attitude error result, Figure 8a–c separately denote the pitch, roll and heading errors. The blue and red line respectively denote the attitude calculated with and without sensor error compensation. Table 12 lists the error statistical result of the two groups attitude result and illustrates that the attitude result with error compensation has an overall better performance. 

As shown in Figure 8, in the first 8 s during which the IMU keeps static, the attitude errors of the two groups are approximately same and almost equal to zero. Then when the IMU begins to rotate, the attitude error curves become different and it is obvious that the attitude with error compensation performed well than that without error compensation. 

During the static period, due to the bias error has been well estimated and removed according to the gyroscope output average, the performances of two groups attitude are nearly same. As analyzed in the MEMS sensor error source section, the scale factor and non-orthogonal errors are proportional to the angular velocity. Therefore, during the rotating stage, these two errors begin to affect the calculation result, and the inertial data with error compensation constrains the negative effect and leads to a more reliable solution. While the sensor data without error compensation suffers from the error effect and the attitude calculated by this group data will drift faster and has a poor performance.

The proposed calibration algorithm is implemented through Kalman filter. The dynamic and measurement model are all expressed in linear form, the state and measurement vector dimension are separately 9 and 3 which is not large; thus, the approach’s complexity is O(9) and does not occupy much computation resource. In addition, sequential measurement update can be applied in filter to avoid calculating the inverse of matrix. Hence, the algorithm only needs limited computing resources and can be embedded in sensor’s internal processor to implement error calibration without support of external CPU. 

## 6. Conclusions

This paper presents a real-time low-cost MEMS gyroscope calibration method without external equipment. This calibration algorithm is able to identify the sensor bias, scale factor error and non-orthogonal error, and the calibration experiment can be easily accomplished by user hand-hold rotation. The system observability is analyzed, and the observable condition of sensor error is investigated. This research can guide the calibration scheme design to save time and achieve reliable estimation result. Both simulated and practical experiments have demonstrated the validation of proposed calibration approach.

## Figures and Tables

**Figure 1 sensors-20-05430-f001:**
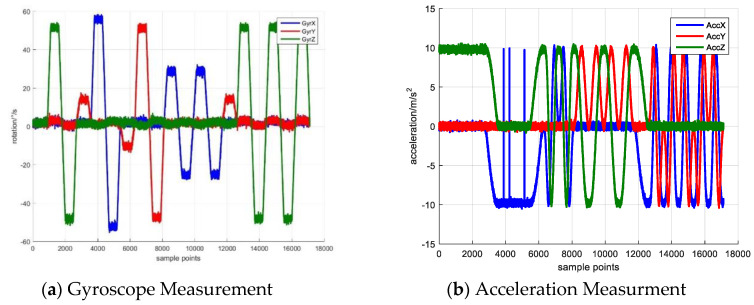
Simulated Inertial Sensor Measurement.

**Figure 2 sensors-20-05430-f002:**
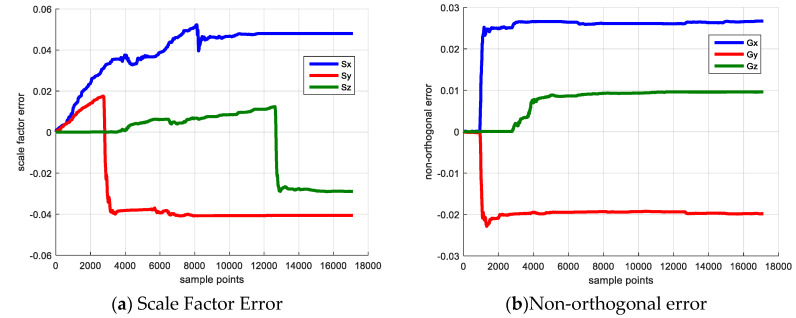
Sensor Error Calibration Result.

**Figure 3 sensors-20-05430-f003:**
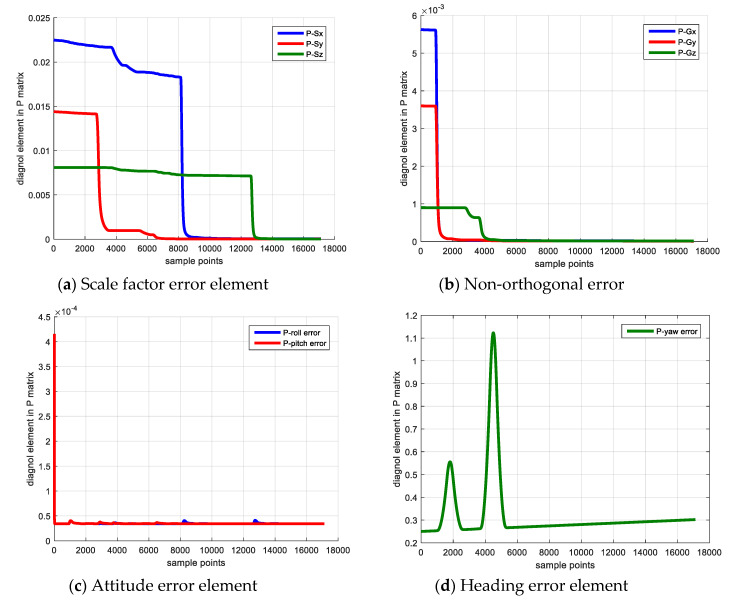
Elements of P matrix.

**Figure 4 sensors-20-05430-f004:**
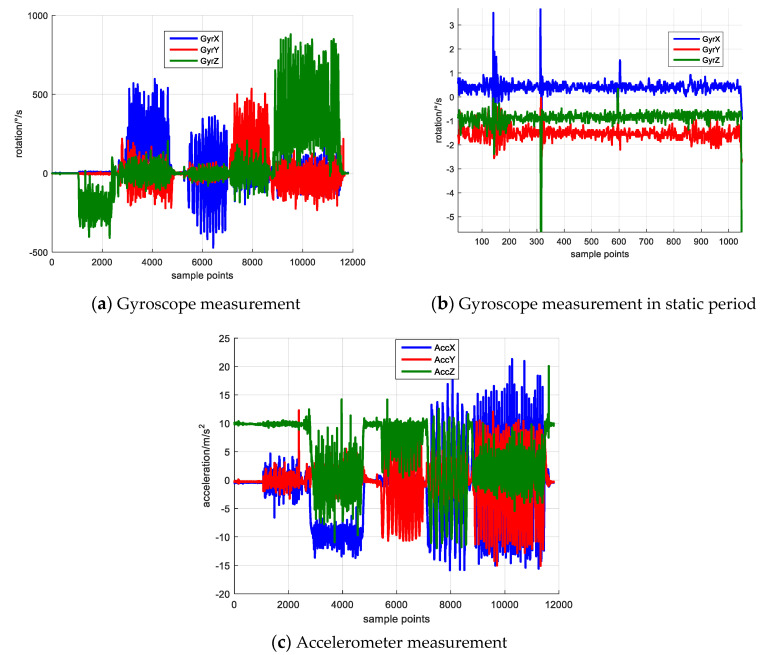
Collected Inertial Data.

**Figure 5 sensors-20-05430-f005:**
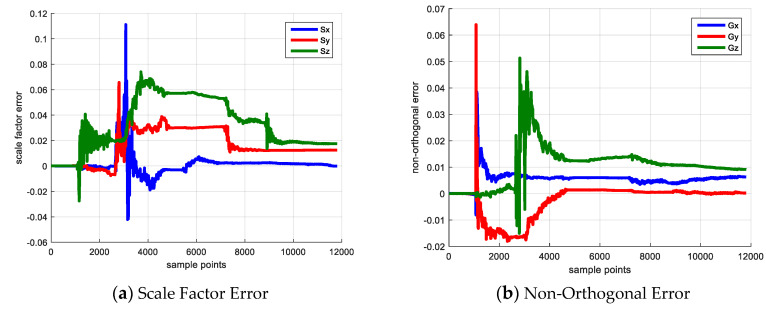
Sensor Error Estimation.

**Figure 6 sensors-20-05430-f006:**
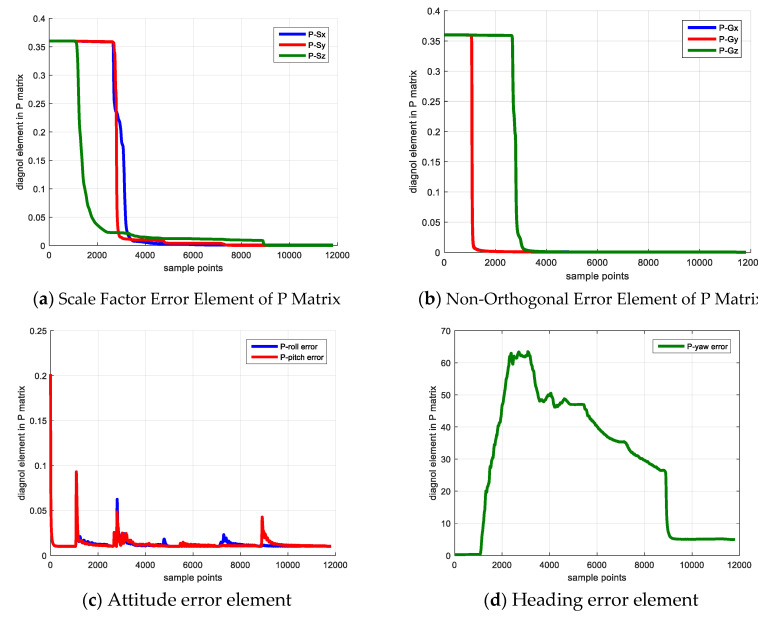
Elements of P Matrix.

**Figure 7 sensors-20-05430-f007:**
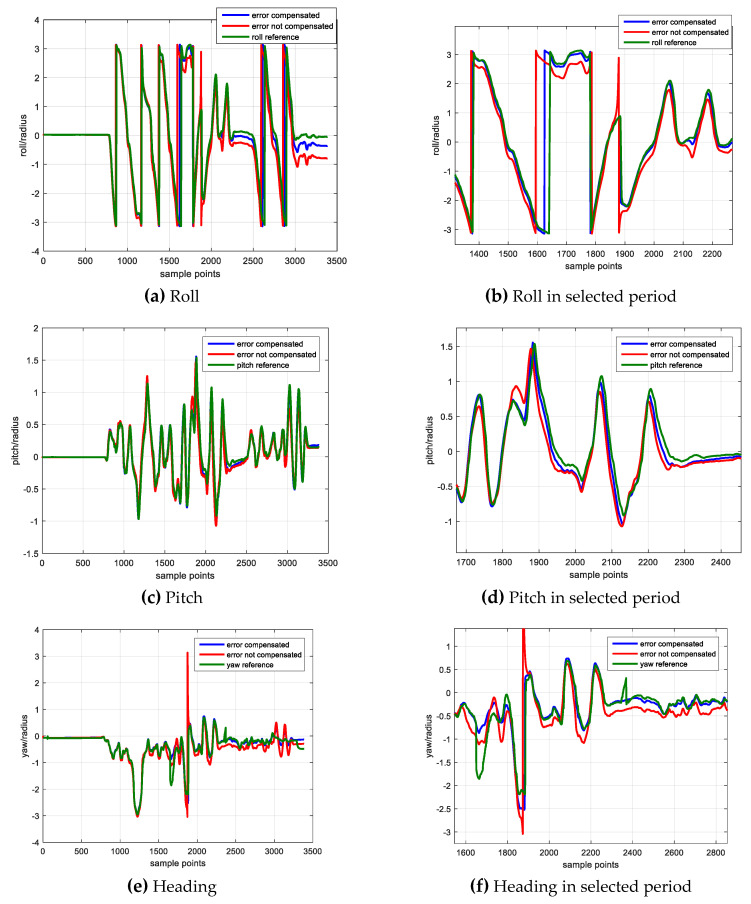
Comparison of Attitude Result.

**Figure 8 sensors-20-05430-f008:**
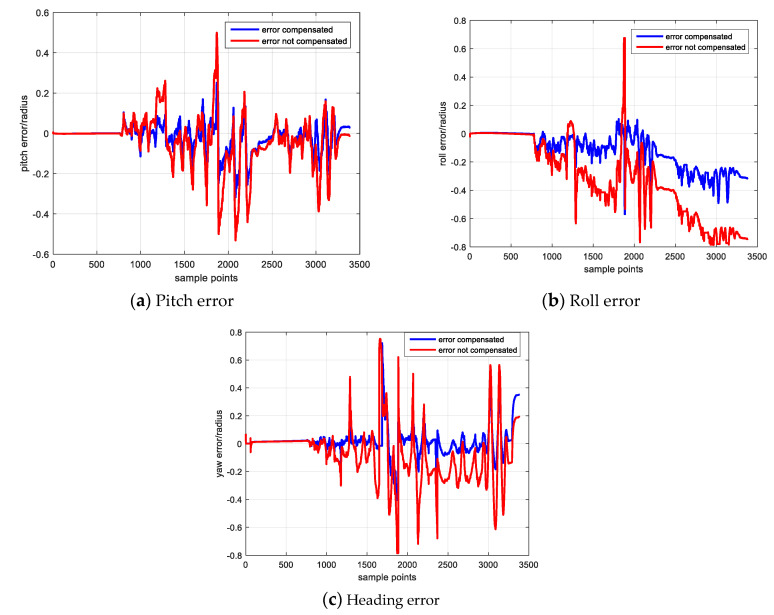
Attitude Calculation Result.

**Table 1 sensors-20-05430-t001:** Difference between Proposed Algorithm and Related Work.

	Bias	Scale Factor Error	Non-Orthogonality	Duration	Calibration Scheme
Fong 2008	√	√	√	30–60 min	×
You [9] 2012	√	√	×	3 min	Empirical suggestion
Li [10] 2015	√	×	×	/	×
Farhad [13] 2020	√	√	√	/	Rotation table
Proposed 2020	√	√	√	3–5 min	√

**Table 2 sensors-20-05430-t002:** MPU-9255/9150 Characteristic.

Parameter	Performance
Bias	±5°/s/±20°/s
Scale factor error	±3%
Non-orthogonal error	±2%
Nonlinearity	±0.1%
ARW	$0.01{}^{\circ}/s/\sqrt{\mathrm{Hz}}$ $0.005{}^{\circ}/s/\sqrt{\mathrm{Hz}}$
Sensitivity scale factor variation over temperature	± 4%
ZRO variation over Temperature	±30°/s/±20°/s −40 °C to + 85 °C

**Table 3 sensors-20-05430-t003:** ADI-16367 Characteristic.

Parameter	Performance
Bias	±3°/s
Scale factor error	±1%
Non-orthogonal error	±1%
Nonlinearity	±0.1%
ARW	$2{}^{\circ}/\sqrt{\mathrm{hr}}$
Sensitivity scale factor variation over temperature	±40 ppm/°C
ZRO variation over Temperature	±0.01°/s −40 °C to + 85 °C

**Table 4 sensors-20-05430-t004:** Crossbow IMU440 Characteristic.

Parameter	Performance
Bias	±1°/s
Scale factor error	±1%
Non-orthogonal error	Not provided
Nonlinearity	±0.5%
ARW	$\;4.5{}^{\circ}/\sqrt{\mathrm{hr}}$
ZRO variation over Temperature	±0.2°/s −40 °C to + 85 °C

**Table 5 sensors-20-05430-t005:** Observability Summarization.

Observable States	Pitch and Roll Angles	Rotating Axis	Rotation
$G_{x},G_{y}$	θ=0,γ=0	*z*-axis (Vertical)	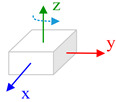
$G_{x},G_{z}$	θ=0,γ=± 90°	*y*-axis (Vertical)	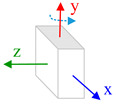
$G_{y},G_{z}$	γ=0,θ=± 90°	*x*-axis (Vertical)	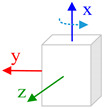
**Combination** ($S_{y},G_{x},G_{z}$)	θ=0	*y*-axis (Horizontal)	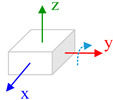
**Combination** ($S_{x},G_{y},G_{z}$)	θ=0	*x*-axis (Horizontal)	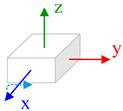
**Combination** ($S_{z},G_{x},G_{y}$)	γ=± 90°	*z*-axis (Horizontal)	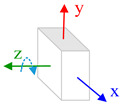

**Table 6 sensors-20-05430-t006:** Simulation Rotation Scheme.

Duration	Rotation	Axis	Direction
0 s–9 s	Static	None	None
9 s–27 s	2π 9 s–18 s −2π 18 s–27 s	*z*	Vertical
27 s–36 s	π/2	y	Switch
36 s–54 s	2π 36 s–45 s −2π 45 s–54 s	*x*	Vertical
54 s–63 s	−π/2	*y*	Switch
63 s–81 s	2π 63 s–72 s −2π 72 s–81 s	*y*	Horizontal
81 s–117 s	π 81 s–90 s 99 s–108 s −π 90 s–99 s 108s–117 s	*x*	Horizontal
117 s–126 s	π/2	*y*	Switch
126 s–171 s	2π 36 s–45 s −2π 45 s–54 s	*z*	Horizontal

**Table 7 sensors-20-05430-t007:** Simulated Error Parameters.

Bias	Scale Factor Error	Non-Orthogonal Error	ARW	VRW
2°/s	$S_{x}=0.05$	$G_{x}=0.025$	$15\;\mathrm{mg}/\sqrt{\mathrm{Hz}}$	$0.5{}^{\circ}/s/\sqrt{\mathrm{Hz}}$
2°/s	$S_{y}=-0.04$	$G_{y}=-0.02$	$15\;\mathrm{mg}/\sqrt{\mathrm{Hz}}$	$0.5{}^{\circ}/s/\sqrt{\mathrm{Hz}}$
2°/s	$S_{z}=-0.03$	$S_{z}=0.01$	$15\;\mathrm{mg}/\sqrt{\mathrm{Hz}}$	$0.5{}^{\circ}/s/\sqrt{\mathrm{Hz}}$

**Table 8 sensors-20-05430-t008:** Bias Calibration Result.

Bias Error	Absolute Error	Relative Error
$B_{x}=1.95{}^{\circ}/s$	0.05	2.5%
$B_{y}=1.96{}^{\circ}/s$	0.04	2.0%
$B_{z}=1.94{}^{\circ}/s$	0.06	3.0 %

**Table 9 sensors-20-05430-t009:** Sensor Error Calibration Result.

Scale Factor Error	Absolute Error	Relative Error	Non-Orthogonal Error	Absolute Error	Relative Error
$S_{x}=0.0480$	0.00196	3.92%	$G_{x}=0.0267$	0.0017	6.8%
$S_{y}=-0.0405$	0.0005	1.25%	$G_{y}=-0.019$	0.0002	1%
$S_{z}=-0.0289$	0.0011	3.67%	$G_{z}=-0.009$	0.0004	4%

**Table 10 sensors-20-05430-t010:** SVD of Non-orthogonal Error.

Rotate along *z*-Axis Vertically		Rotate along *x*-Axis Vertically	
Full State	Without *G_x_*, *G_y_*	Full State	Without *G_y_*, *G_z_*
9.8	9.8	9.8	9.8
9.8	9.8	9.8	9.8
8.524092	0.61935	9.23949	0.395472
8.50978	0.375294	9.23464	0.255418

**Table 11 sensors-20-05430-t011:** SVD of Scale Factor Error.

Rotate along *x*-Axis Horizontally		Rotate along *y*-Axis Horizontally		Rotate along *z*-Axis Horizontally	
Full State	Without *S_x_*, *G_y_*, *G_z_*	Full State	Without *S_y_*, *G_x_*, *G_z_*	Full State	Without *S_z_*, *G_x_*, *G_y_*
9.8	9.8	9.8	9.8	9.8	9.8
9.8	9.8	9.8	9.8	9.8	9.8
4.792952	0.525420	8.065646	0.236580	8.096383	0.715018
4.644314	0.000069	8.017878	0.000045	7.994556	0.000171

**Table 12 sensors-20-05430-t012:** Statistical Error Result.

	Error without Compensation		Error with Compensation	
	Mean	RMS	Mean	RMS
Pitch	−0.0362	0.1316	−0.0174	0.0688
Roll	0.30975	0.4141	−0.1210	0.1751
Heading	0.07288	0.2139	0.01846	0.1104

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
