# Peer review of "A Novel MEMS Gyroscope In-Self Calibration Approach"

_sensors, 2020, doi:10.3390/s20185430_

Round 1
Reviewer 1 Report
The paper is clear and well written. While the approach is interesting, there are a couple of issues that deserve some discussion:
- It is well known that thermal drifts are very important (it is noted in the paper introduction), but the proposed algorithm and method does not take into account this. Then, what it is the scope of usability of the proposed method?
- I suggest adding the following reference and discussing the relation with the proposed method
Temperature-Dependent Calibration of Triaxial Sensors: Algorithm, Prototype, and Some Results. P Bernal-Polo, H Martínez-Barberá. IEEE Sensors Journal 20 (2), 876-884
- In section 3.3.1 dynamic model the bias is calibrated by averaging. How good is the repeatability of this measure? That is, if this average is computed several times at different times in different experiments
- The method relies in the computation of the gravity vector using the accelerometers. The introduction states "pre-calibrated". How is this done? The low cost accelerometers are known to be very noisy. How this affects the derived gravity vector?
Related to the experiments, I understand that it is assumed that no variation of temperature occurs, which can be assumed after some warm up time. It might be worth noting. There are a couple of issues/suggestions:
- The experiments are developed with the MPU-9150, but its characteristics are not presented, for instance in section 2, where MPU-9255, ADI-16367 and IMU440 do.
- The practical experiment mimics the simulation. How is the timing and alignment satisfied with respect to the simulation? It is needed any kind of testing device (like the one presented in the reference above)? If so, could you specify or describe?
Author Response
Dear reviewer,
Thank you for providing us the impressive comments and suggestions. These comments are not only useful for improving the quality of this paper, but also valuable for our research and writing in the future. We have answered all the questions and made corresponding revisions in the manuscript. We hope these changes are useful to make this paper more qualified.
The requests for the questions and suggestions are listed as below:
- It is well known that thermal drifts are very important (it is noted in the paper introduction), but the proposed algorithm and method does not take into account this. Then, what it is the scope of usability of the proposed method?
Yes. The thermal drifts are very important and the temperature has large effect on the gyroscope measurement performance. We did not consider the temperature issue as the calibration process can be accomplished in 3-5 minutes and the temperature varies in very limited.
Thanks for point this out.
The scope of usability of proposed method is:
- The pre-calibrated accelerometer (i.e., bias error and scale factor error have been removed) is required to be used in measurement model of Kalman filter.
- We assume that the temperature keeps static during calibration process and as such the gyroscope bias error does not vary.
- I suggest adding the following reference and discussing the relation with the proposed method. Temperature-Dependent Calibration of Triaxial Sensors: Algorithm, Prototype, and Some Results. P Bernal-Polo, H Martínez-Barberá. IEEE Sensors Journal 20 (2), 876-884
Thanks for the suggestion!
The mentioned reference has been cited in revised paper and the following discussion of relation has been added between Line 85-90.
Pablo[14] introduces a temperature-dependent calibration algorithm to calibrate triaxial sensors and develops an electromechanical prototype to perform the approach. Compared with this algorithm, our method does not need to consider the temperature element as the whole calibration procedure can be accomplished in short time without temperature variation and as such gyroscope bias error does not vary. Additionally, any other testing device is unnecessary.
- In section 3.3.1 dynamic model the bias is calibrated by averaging. How good is the repeatability of this measure? That is, if this average is computed several times at different times in different experiments
In each experiment, we put the IMU in static for approximately 30 seconds, and then start to rotate it according to designed scheme as described in Section Test and Result. The gyroscope output during static stage is averaged to calculate as its bias. The sensor bias has no variation during this short procedure; thus, we do not have to compute several times at different times in different experiments and consider its repeatability.
In practical application, the gyroscope bias is also calibrated by averaging its output in static period; however, it may vary due to various elements and cannot have good repeatability (suffer from drift and the bias does not equal to its pre-calibrated value). Generally, the gyroscope bias is modelled in Kalman filter as first-order Markov process to be estimated with measurement update. The specific operation can refer to equation (9) in paper, the gyroscope bias error will be considered as state vector, and the scale factor and non-orthogonal error is not modelled in filter as they have been calibrated before usage and are not environment dependent parameters.
- The method relies in the computation of the gravity vector using the accelerometers. The introduction states "pre-calibrated". How is this done? The low-cost accelerometers are known to be very noisy. How this affects the derived gravity vector?
Ref[19] introduces an accelerometer calibration method which does not require any external calibration faculties, and we make use of this method to calibrate accelerometer to derive its bias error, scale factor error and non-orthogonal error.
The main principle of this calibration method is to use the fact that regardless of the direction that the IMU axes are pointing, the total values sensed by the accelerometers in static mode should be equal to the gravity. To perform the accelerometer calibration, its data measured in different attitude is collected (more orientation variation is better), and the nonlinear Least-Square algorithm is applied to identify the error parameter. The specific mathematical model, algorithm design have been well illustrated in Ref[19].
Regarding to the noise of accelerometer:
- The accelerometer noise density is applied to set R matrix element in Kalman filter to balance its contribution in measurement update.
- Currently, with the development of MEMS sensor manufacturing, the random noise can be well constrained. According to MPU series datasheet, the accelerometer noise power spectral density is 300ug/sqrt(Hz)
- We cloud decrease the sensor’s sampling frequency and set suitable low-pass filter cut-off frequency to best fit our application and avoid the effect of high- frequency noise.
- Related to the experiments, I understand that it is assumed that no variation of temperature occurs, which can be assumed after some warm up time. It might be worth noting. There are a couple of issues/suggestions:
The experiments are developed with the MPU-9150, but its characteristics are not presented, for instance in section 2, where MPU-9255, ADI-16367 and IMU440 do.
The practical experiment mimics the simulation. How is the timing and alignment satisfied with respect to the simulation? It is needed any kind of testing device (like the one presented in the reference above)? If so, could you specify or describe?
Thanks for the suggestion.
- Thanks for pointing this sensor type issue, the MPU-9150 error characteristics has been added to Table 1
- The experiment test is accomplished totally within 3 minutes. Due to the difference between various complex elements existed in practice and ideal environment in simulation, the result in these two experiments cannot perfectly aligned. However, as shown in Figure 6 in paper, the scale factor error, non-orthogonal error and their corresponding elements in P matrix became well converged after special dynamic rotation has been accomplished which accords to observability analysis in paper. Moreover, the sensor error estimation result in its reasonable range as listed in datasheet and the attitude result with and without error compensation can show its validation.
- Our IMU is a wireless one, and does not need electrical cable for power supply and data transmitted; thus, it is convenient for us to rotate in hand to collect data without any constrain. The sensor raw measurement can be wirelessly sent to a receiver connected to PC. Then the collected data is used to test the algorithm. Hence, we do not need a testing device as illustrated in above reference. The following figure shows our wireless IMU.

Reviewer 2 Report
In this manuscript, the authors claim “A novel MEMS Gyroscope In-Self Calibration Approach.” The proposed method does not need the support of external high-precision equipment. A Kalman filter is design to perform the calibration procedure and estimate gyroscope bias error, scale factor error and non-orthogonal error. Both simulated and practical experiments are carried out to test the validation of the proposed calibration algorithm. However, there are several points the authors need to address: 1. Why the ARW of MPU-9255 is smaller than ADI-16367 and IMU440 in table 2 to 4? 2. In Table 2 to 4, the bias parameters are given, I want to know how to get this parameters? 3. In eq.(3), the authors suggest the averaging operation for bias calculation, it is well-known that the Earth rate is also coupling in the outputs of gyroscopes, how to eliminate this angular rate? 4. In the manuscript, the authors use the accelerometers to calibrated the gyroscopes. Does the errors of the accelerometers are considered? It is also known that the errors of the consumer-accelerometers are large. All these influences of the errors of the accelerometers should be considered. The misalignments between accelerometers and gyroscopes are also need to given. 5. In figure 7, how to get the reference information? 6. In figure 8 (c), why the error of the compensated gyroscopes is larger than the uncompensated gyroscopes in the last of the sampling points?Author Response
Dear reviewer,
Thank you for providing us so many impressive comments and suggestions. These comments are not only useful for improving the quality of this paper, but also valuable for our research and writing in the future. We have answered all the questions and made corresponding revisions in the manuscript. We hope these changes are useful to make this paper more qualified.
The requests for the questions and suggestions are listed as below:
- Why the ARW of MPU-9255 is smaller than ADI-16367 and IMU440 in table 2 to 4?
The ARW value of each sensor is quoted from sensor datasheet. I cannot know the exact reason why the MPU-9255 sensor has the least ARW as I mainly focus on the IMU sensor’s application and owns limited knowledge in its internal measurement principle and sensor manufacturing; however, based on my best knowledge,
The reason may be the high-level IMU sensor cannot have an overall better performance and it has performed better in bias, orthogonality, maintaining good measurement with variation of temperature.
In addition, the main drawbacks of consumer level IMU include: 1) suffers from large bias error and bias instability which means that the bias may vary with time or turning on-off process; 2) is easily affected by temperature. For the consumer-level IMU, MPU-9255, in the temperature range of -40 °C to +85°C, the sensitivity scale factor variation is 4%, and the variation of Zero Rate Output (ZRO) is up to 30°/s. The scale factor error change of ADI1367 over temperature is 40ppm/°C, and the bias variation with temperature is 0.01°/s/°C.
- In Table 2 to 4, the bias parameters are given, I want to know how to get this parameters?
The sensor biases errors are all quoted from the sensor datasheet listed in reference [15-18] of paper.
The sensor bias parameter of each sensor is highlighted in read rectangle in following figures for reference. One point to mention is that, actually the ADI 16367 bias error is 3 degree/second instead of 1 degree/second depicted in peer review paper. Thanks for the reviewer to remind us re-exam the sensor error characteristic and find this typo.
- In eq.(3), the authors suggest the averaging operation for bias calculation, it is well-known that the Earth rate is also coupling in the outputs of gyroscopes, how to eliminate this angular rate?
The earth rotation is 15 degree/hour and equals to 0.0042 degree/second. Compared with the least bias error 1 degree/second among the three IMUs, the bias error is 200 times of the Earth rotation. Hence, the earth rotation is much less than the sensor error and can be safely ignored.
The content between 211-216 has described the earth rotation issue as well.
- In the manuscript, the authors use the accelerometers to calibrated the gyroscopes. Does the errors of the accelerometers are considered? It is also known that the errors of the consumer-accelerometers are large. All these influences of the errors of the accelerometers should be considered. The misalignments between accelerometers and gyroscopes are also need to given.
As illustrated between line 237-239, “It should be noted that the accelerometer has been well calibrated and its biases, scale factor errors and non- orthogonal errors have been removed[21]”. Ref[21] introduces a calibration method making use of the fact that regardless of the direction that the IMU axes are pointing, the total values sensed by the accelerometers in static mode should be equal to the gravity. The accelerometer calibration method does not need the support of external equipment and only needs to put IMU with various orientation in static.
Generally, the accelerometer measurement data in different attitudes is collected for calibration and the nonlinear Least-Square algorithm is applied to identify the error parameter.
The misalignment between accelerometers and gyroscopes has been considered and added to equation (10)
- In figure 7, how to get the reference information?
The reference attitude is derived from the commercial Attitude calculation software embedded in SmartPhone Framework.
- In figure 8 (c), why the error of the compensated gyroscopes is larger than the uncompensated gyroscopes in the last of the sampling points?
Based on my best knowledge, the element two negatives makes a positive coupled together may lead to this phenomenon.
However, the statistical error results and error plot figures have shown that the error with compensation will derive overall much better performance.

Reviewer 3 Report
The work describes gyroscope calibration using manual rotation and well calibrated accelerometer. The method may be useful for calibrating low cost consumer grade devices.
The manuscript should be improved in detailing how data is analyzed and error plots generated. Is this implemented for example in C, python, or Matlab? Furthermore, the method becomes more interesting if it can be implemented in smart sensors without external CPU. Some discussion about this possibility/feasibility would enhance the manuscript.
Author Response
Dear reviewer,
Thank you for providing us the impressive comments and suggestions. These comments are not only useful for improving the quality of this paper, but also valuable for our research and writing in the future. We have answered all the questions and made corresponding revisions in the manuscript. We hope these changes are useful to make this paper more qualified.
The requests for the questions and suggestions are listed as below:
- The manuscript should be improved in detailing how data is analyzed and error plots generated. Is this implemented for example in C, python, or Matlab?
We add the following paragraph in the beginning of Section “5 Line 365--374. Test and Result”to introduce how the experiment is designed, carried out and how data is analyzed.
“Simulated and practical experiments have been carried out to test and verify the proposed calibration algorithm. In simulated experiment, we first design the sensor rotation sequence to generate true sensor data according to observability analysis and add the preset sensor error to data which is used to perform the calibration approach. In practical experiment, the rotation follows the same sequence as that performed in simulation. In each experiment, the converge of estimated scale factor error and non-orthogonal error and their corresponding element in P matrix are drawn, compared and analyzed. In simulation experiment, we compare the estimated sensor error and preset error parameter to derive the absolute error and relative error. In practical experiment, the attitude calculated by sensor data with and without error compensation are compared to demonstrate the validation of calibrated parameter and all the figures are plotted in Matlab.”
- Furthermore, the method becomes more interesting if it can be implemented in smart sensors without external CPU. Some discussion about this possibility/feasibility would enhance the manuscript.
We add the following content as the last paragraph in Section “5 Line 530-536. Test and Result”to further discuss the possibility of the algorithm’s application in smart sensors.
The proposed calibration algorithm is implemented through Kalman filter. The dynamic and measurement model are all expressed in linear form, the state and measurement vector dimension are separately 9 and 3 which is not large; thus, the approach’s complexity is O(9) and does not occupy much computation resource. In addition, sequential measurement update can be applied in filter to avoid calculating the inverse of matrix. Hence, the algorithm only needs limited computing resources and can be embedded in sensor’s internal processor to implement error calibration without support of external CPU.

Reviewer 4 Report
The manuscript describes a hand-held in-self calibration method for MEMS gyroscope using Kalman filter. The scale factor error, non-orthogonal error and bias error are all estimated using hand rotation. The paper also provides design principles of optimal calibration. The data from simulation and experiments provided strong evidence for validation of the proposed method. The calibration method would eliminate the use of external high precision equipment and is a promising solution for calibration of low cost MEMS gyroscope.
Overall, this is a well written paper. The method is interesting and the conclusion is well supported by the data. How would the authors determine the accuracy of the axes x, y, z in their hand calibration? I would suggest the authors to give discussion on cross-talk errors.
Author Response
Dear reviewer,
Thank you for providing us the impressive comments and suggestions. These comments are not only useful for improving the quality of this paper, but also valuable for our research and writing in the future. We have answered all the questions and made corresponding revisions in the manuscript. We hope these changes are useful to make this paper more qualified.
The requests for the questions and suggestions are listed as below:
- The method is interesting and the conclusion is well supported by the data. How would the authors determine the accuracy of the axes x, y, z in their hand calibration? I would suggest the authors to give discussion on cross-talk errors.
Thanks for your suggestion!
- In simulation experiment, we compare the estimated sensor error and preset error parameter to derive the absolute error and relative error. Table 8,9 list the quantitative error characteristic. In practical experiment, we compare the attitude derived through sensor measurement with and without estimated error compensation. The result shows that the attitude derived through data with error compensation has much better performance. Figure 7,8 and Table 10 show and list the result
- Regarding to cross-talk error, Non-orthogonal error can be considered as one type of cross-talk error. As the name suggests, non-orthogonal errors occur when any of the axes of the sensor triad depart from mutual orthogonality. This usually happens at the time of manufacturing. The figure below depicts the case of the z-axis being misaligned by an angular offset of from xz-plane and from the yz-plane.
Figure 1: Nonorthogonality of z-axis to xy plane
The proposed calibration algorithm is to estimate the gyroscope sensor error including non-orthogonal error and scale factor error to derive more accurate rotation measurement.
